# A Neural Knowledge Language Model

**Sungjin Ahn**[1], **Heeyoul Choi**[2,*] **Tanel Pärnamaa**[3], **& Yoshua Bengio**[4]
[1,3,4]Université de Montréal, [2]Handong Global University, [4]CIFAR Senior Fellow
{[1]sjn.ahn, [2]heeyoul,[3]tanel.parnamaa}@gmail.com
{[4]yoshua.bengio}@umontreal.ca

## Abstract

Current language models have significant limitations in their ability to encode and decode factual knowledge. This is mainly because they acquire such knowledge based on statistical co-occurrences, even if most of the knowledge words are rarely observed named entities. In this paper, we propose a Neural Knowledge Language Model (NKLM) which combines symbolic knowledge provided by a knowledge graph with the RNN language model. The model predicts whether the word to generate has an underlying fact or not. Then, a word is either generated from the vocabulary or copied from the description of the predicted fact. We train and test the model on a new dataset, *WikiFacts*. In experiments, we show that the NKLM significantly improves the perplexity while generating a much smaller number of unknown words. In addition, we demonstrate that the sampled descriptions include named entities which used to be the unknown words in RNN language models.

## 1 Introduction

> *Kanye West, a famous <unknown> and the husband of <unknown>,*
> *released his latest album <unknown> in <unknown>.*

A core purpose of language is to communicate knowledge. Thus, for human-level language understanding, it is important for a language model to take advantage of knowledge. Although traditional language models are good at capturing statistical co-occurrences of entities as long as they are observed frequently in a corpus (e.g., words like verbs, pronouns, and prepositions), they are in general limited in their ability to encode or decode knowledge, which is often represented by named entities such as person names, place names, years, etc. (as shown in the above example sentence of Kanye West.) When trained with a very large corpus, traditional language models have demonstrated to some extent the ability to encode/decode knowledge (Vinyals & Le, 2015; Serban et al., 2015). However, we claim that simply feeding a larger corpus into a bigger model hardly results in a good knowledge language model.

The primary reason for this is the difficulty in learning good representations for rare or unknown words because these are a majority of the knowledge-related words. In particular, for applications such as question answering (Iyyer et al., 2014; Weston et al., 2016; Bordes et al., 2015) and dialogue modeling (Vinyals & Le, 2015; Serban et al., 2015), these words are of our main interest. Specifically, in the recurrent neural network language model (RNNLM) (Mikolov et al., 2010) the computational complexity is linearly dependent on the number of vocabulary words. Thus, including all words of a language is computationally prohibitive. Instead, we typically fill our vocabulary with a limited number of frequent words and regard all the other words as the unknown (UNK) word. Even if we can include a large number of words in the vocabulary, according to Zipf's law, a large portion of the words will be rarely observed in the corpus and thus learning good representations for these words remains a problem.

The fact that languages and knowledge can change over time also makes it difficult to simply rely on a large corpus. Media produce an endless stream of new knowledge every day (e.g., the results of baseball games played yesterday) that is even changing over time (e.g., "*the current president of the*

---

*This work was done while HC was in Samsung Advanced Institute of Technology

*United States is ___"*). Furthermore, a good language model should exercise some level of reasoning. For example, it may be possible to observe several occurrences of Barack Obama's year of birth in a large corpus and thus the model may be able to predict it. However, after seeing mentions of his year of birth, presented with a simple reformulation of that piece of knowledge into a sentence such as *"Barack Obama's age is ___"*, one would not expect current language models to handle the required amount of reasoning in order to predict the next word (i.e. the age) easily. However, a good model should be able to reason the answer from this context[1].

In this paper, we propose a Neural Knowledge Language Model (NKLM) as a step towards addressing the limitations of traditional language modeling when it comes to exploiting factual knowledge. In particular, we incorporate symbolic knowledge provided by a knowledge graph (Nickel et al., 2015) into the RNNLM. A knowledge graph (KG) is a collection of facts which have a form of *(subject, relationship, object)*. We observe particularly the following properties of KGs that make the connection to the language model sensible. First, facts in KGs are mostly about rare words in text corpora. KGs are managed and updated in a similar way that Wikipedia pages are managed to date. The KG embedding methods (Bordes et al., 2011; 2013) provide distributed representations for the entities in the KG. The graph can be traversed for reasoning (Gu et al., 2015). Finally, facts come along with textual representations which we call the *fact description* and take advantage of here.

There are a few differences between the NKLM and the traditional RNNLM. First, we assume that a word generation is either based on a fact or not. Thus, at each time step, before predicting a word, we predict whether the word to generate has an underlying fact or not. As a result, our model provides the predictions over facts in a topic in addition to the word predictions. Similarly to how context information of previous words flows through the hidden states in the RNNLM, in the NKLM the previous information on both facts and words flow through an RNN and provide richer context. Second, the model has two ways to generate the next word. One option is to generate a "vocabulary word" from the vocabulary softmax as is in the RNNLM. The other option is to generate a "knowledge word" by *copying* a word contained in the description of the predicted fact. Considering that the fact description is often short and consists of out-of-vocabulary words, we predict the position of the word to copy within the fact description. This *knowledge-copy* mechanism makes it possible to generate words which are not in the predefined vocabulary. Thus, it does not require to learn explicit embeddings of the words to generate, and consequently resolves the rare/unknown word problem. Lastly, the NKLM can immediately adapt to adding or modifying knowledge because the model learns to predict facts, which can easily be modified without having to retrain the model.

Training the above model in a supervised way requires to align words with facts. To this end, we introduce a new dataset, called *WikiFacts*. For each topic in the dataset, a set of facts from the Freebase KG (Bollacker et al., 2008) and a Wikipedia description of the same topic is provided along with the alignment information. This alignment is done automatically by performing string matching between the fact description and the Wikipedia description.

## 2 RELATED WORK

There have been remarkable advances in language modeling research based on neural networks (Bengio et al., 2003; Mikolov et al., 2010). In particular, the RNNLMs are interesting for their ability to take advantage of longer-term temporal dependencies without a strong conditional independence assumption. It is especially noteworthy that the RNNLM using the Long Short-Term Memory (LSTM) (Hochreiter & Schmidhuber, 1997) has recently advanced to the level of outperforming carefully-tuned traditional n-gram based language models (Jozefowicz et al., 2016).

There have been many efforts to speed up the language models so that they can cover a larger vocabulary. These methods approximate the softmax output using hierarchical softmax (Morin & Bengio, 2005; Mnih & Hinton, 2009), importance sampling (Jean et al., 2015), noise contrastive estimation (Mnih & Teh, 2012), etc. Although helpful to mitigate the computational problem, these approaches still suffer from the statistical problem due to rare or unknown words. Having the UNK word as the output of a generative language model is also inconvenient (e.g, dialogue system).

---

[1]We do not investigate the reasoning ability in this paper but highlight this example because the explicit representation of facts would help to handle such examples.

To help deal with the rare/unknown word problem, the pointer networks (Vinyals et al., 2015) have been adopted to implement the copy mechanism (Gulcehre et al., 2016; Gu et al., 2016) and applied to machine translation and text summarization. With this approach, the (unknown) word to copy from the context sentence is inferred from neighboring words. However, because in our case the context can be very short and often contains no known relevant words (e.g., person names), we cannot use the existing approach directly.

Our knowledge memory is also related to the recent literature on neural networks with external memory (Bahdanau et al., 2014; Weston et al., 2015; Graves et al., 2014). In Weston et al. (2015), given simple sentences as facts which are stored in the external memory, the question answering task is studied. In fact, the tasks that the knowledge-based language model aims to solve (i.e. predict the next word) can be considered as a fill-in-the-blank type of question answering. The idea of jointly using Wikipedia and knowledge graphs has also been used in the context of enriching word embedding (Celikyilmaz et al., 2015; Long et al., 2016).

# 3 MODEL

## 3.1 PRELIMINARY

A topic[2] $k$ in a set of entities $\mathcal{E}$ is associated with *topic knowledge* $\mathcal{F}_k$ (e.g., from Freebase) and *topic description* $W_k$ (e.g., from Wikipedia). Topic knowledge $\mathcal{F}_k$ is a set of facts $\{a^{k,1}, a^{k,2}, \ldots, a^{k,|\mathcal{F}_k|}\}$ where each fact $a$ is a triple of subject $\in \mathcal{E}$, relationship, and object $\in \mathcal{E}$, e.g., (*Barack Obama, Married-To, Michelle Obama*). Topic description $W_k$ is a sequence of words $(w_1^k, w_2^k, \ldots, w_{|W_k|}^k)$ describing the topic (e.g., a description of a topic in Wikipedia). Because the subject entities in $\mathcal{F}_k$ are all equal to the topic entity $k$[3] and the words describing relationships can easily be found in the vocabulary, we use the description of the object entity (e.g., *Michelle Obama*) as our fact description.

Given $\mathcal{F}_k$ and $W_k$, we perform simple string matching between words in $W_k$ and words in the fact descriptions in $\mathcal{F}_k$ and thereby build a sequence of augmented observations $Y_k = \{y_t^k = (w_t, a_t, z_t)\}_{t=1:|W_k|}$. Here, $w_t \in W_k$ is an observed word, $a_t \in \mathcal{F}_k$ a fact on which the generated word $w_t$ is based, and $z_t$ a binary variable indicating whether $w_t$ is in the vocabulary $\mathcal{V}$ (including UNK) or not. Because not all words are based on a fact (e.g., words like, *is, a, the, have*), we introduce a special type of fact, called Not-a-Fact (NaF), and assign NaF to such words.

For example, a description "*Rogers was born in Latrobe, Pennsylvania in 1928*" from a topic Fred Rogers in Wikipedia, is augmented to, $Y = \{(w=$"Rogers", $a=0$, $z=0$), ("was", NaF, 1), ("born", NaF, 1), ("in", NaF, 1), ("Latrobe", 42, 0), ("Pennsylvania", 42, 1), ("in", NaF, 1), ("1928", 83, 0)}. Here, we use facts on Fred Rogers, $a^{42} = $ (Fred_Rogers, Place_of_Birth, Latrobe_Pennsylvania), $a^{83} = $ (Fred_Rogers, Year_of_Birth, 1928), and a special fact $a^0 = $ (Fred_Rogers, Topic_Itself, Fred_Rogers) which we define in order to refer to the topic string itself. We also assume here that the words *Rogers, Latrobe* and *1928* are not in the vocabulary.

During the inference and training of topic $k$, we assume that the topic knowledge $\mathcal{F}_k$ is loaded in the *knowledge memory* in a form of a matrix $\mathbf{F}_k \in \mathbb{R}^{D_a \times |\mathcal{F}_k|}$ where the $i$-th column is a fact embedding $\mathbf{a}^{k,i} \in \mathbb{R}^{D_a}$. The fact embedding is the concatenation of subject, relationship, and object embeddings. We obtain these entity embeddings from a preliminary run of a knowledge graph embedding method such as TransE (Bordes et al., 2013). Note that we fix the fact embedding during the training of our model to help the model predict new facts at test time. But, we learn the embedding of the Topic_Itself. For notation, to denote the vector representation of any object of our interest, we use bold lowercase characters. For example, the embedding of a word $w_t$ is represented by $\mathbf{w}_t = \mathbf{W}[w_t]$ where $\mathbf{W}^{D_w \times |\mathcal{V}|}$ is the word embedding matrix, and $\mathbf{W}[w_t]$ denotes the $w_t$-th column of $\mathbf{W}$.

---

[2]In this work, a topic is one of the entities which exist in both Wikipedia and Freebase. This is different to the concept in topic modeling where a topic is represented by a distribution over words.

[3]Although in Freebase the topic entity can be either the subject or the object, for convenience we process them such that the subject is always equal to the topic entity $k$.

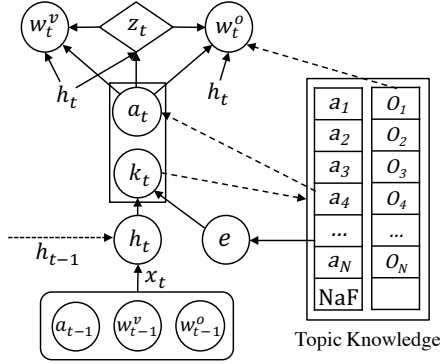

Figure 1: The NKLM model. The input consisting of a word (either $\mathbf{w}_{t-1}^o$ or $\mathbf{w}_{t-1}^v$) and a fact ($\mathbf{a}_{t-1}$) goes into LSTM. The LSTM's output $\mathbf{h}_t$ together with the knowledge context $\mathbf{e}$ generates the fact key $\mathbf{k}_t$. Using the fact key, the fact embedding $\mathbf{a}_t$ is retrieved from the topic knowledge memory. Using $\mathbf{a}_t$ and $\mathbf{h}_t$, knowledge-copy switch $z_t$ is determined, which in turn determines the next word generation source $\mathbf{w}_t^v$ or $\mathbf{w}_t^o$. The copied word $\mathbf{w}_t^o$ is a symbol taken from the fact description $\mathcal{O}_{a_t}$.

Topic Knowledge

## 3.2 INFERENCE

At each time step, the NKLM follows four sub-steps. First, using both the word and fact outputs from the previous time step as the input of the current time step, we update the LSTM controller. Second, given the output of the LSTM, the NKLM predicts a fact (including NaF) and extracts corresponding fact embedding from the knowledge memory. Thirdly, with the extracted fact and the state of the LSTM controller, the NKLM makes a binary decision to choose the source of word generation. Finally, a word is generated according to the chosen source. A model diagram is depicted in Fig. 1. In the following, we describe these four steps in more detail.

**1) Input Representation and LSTM Controller.** As shown in Fig. 1, the input at time step $t$ is the concatenation of three embedding vectors corresponding to a fact $a_{t-1}$, a vocabulary word $w_{t-1}^v$, and a copied word $w_{t-1}^o$, all predicted in the previous time step. However, because at a time step, the predicted word comes only either from the vocabulary or by copying from the fact description, we set either $w_{t-1}^v$ or $w_{t-1}^o$ to a zero vector when it is not selected in the previous step. As we shall see, we use position embeddings to represent the copied words by its position within the fact description. And, because the dimensions of the vocabulary word embedding and the position embedding for copied words are different, we use such concatenation of $w_{t-1}^v$ and $w_{t-1}^o$ to represent the word input. The resulting input representation $\mathbf{x}_t = f_{\text{concat}}(\mathbf{a}_{t-1}, \mathbf{w}_{t-1}^v, \mathbf{w}_{t-1}^o)$ is then fed into the LSTM controller, and obtain the output states $(\mathbf{h}_t, \mathbf{c}_t) = f_{\text{LSTM}}(\mathbf{x}_t, \mathbf{h}_{t-1})$. Note that $\mathbf{a}_{t-1}$ and $\mathbf{w}_{t-1}^o$ (e.g., corresponding to $n$-th position) together can deliver information that a symbol in $n$-th position in the description of fact $a_{t-1}$ was used in the previous time step.

**2) Fact Extraction.** Then, we predict a relevant fact $a_t$ on which the word $w_t$ will be based. If the word $w_t$ is supposed to be irrelevant to any fact, the NaF type is predicted. Unlike the fact embeddings, we learn the NaF embedding during training.

Predicting a fact is done in two steps. First, a fact-key $\mathbf{k}_{\text{fact}} \in \mathbb{R}^{D_a}$ is generated by $\mathbf{k}_{\text{fact}} = f_{\text{factkey}}(\mathbf{h}_t, \mathbf{e}_k)$. Here, $\mathbf{e}_k \in \mathbb{R}^{D_a}$ is the topic context embedding (or a subgraph embedding of the topic) which encodes information about what facts are available in the knowledge memory so that the key generator adapts to changes in the knowledge memory. For example, if we remove a fact from the memory, *without retraining*, the fact-key generator should be aware of the absence of that information and thus should not generate a key vector for the removed fact. Although, in the experiments, we use mean-pooling (average of the all fact embeddings in the knowledge memory) to obtain $\mathbf{e}_k$, one can also consider using the soft-attention mechanism (Bahdanau et al., 2014). For the fact-key generator $f_{\text{factkey}}$, we use an MLP with one hidden layer of ReLU nonlinearity.

Then, using the generated fact-key $\mathbf{k}_{\text{fact}}$, we perform key-value lookup over the knowledge memory $\mathbf{F}_k$ to predict a fact and retrieve its embedding $\mathbf{a}_t$,

$$P(a_t|h_t) = \frac{\exp(\mathbf{k}_{\text{fact}}^\top \mathbf{F}_k[a_t])}{\sum_{a'} \exp(\mathbf{k}_{\text{fact}}^\top \mathbf{F}_k[a'])}, \tag{1}$$

$$a_t = \underset{a_t \in \mathcal{F}_k}{\arg\max}\, P(a_t|h_t), \tag{2}$$

$$\mathbf{a}_t = F_k[a_t]. \tag{3}$$

Note that in order to perform the copy mechanism, we need to pick a single fact from the knowledge memory instead of using the weighted average of the fact embeddings as in the soft-attention.

**3) Knowledge-Copy Switch.** Given the encoding of the context $\mathbf{h}_t$ and the embedding of the extracted fact $\mathbf{a}_t$, the model decides the source for the next word generation: either from the vocabulary or from the fact description by copy. As $z_t = 1$ if the word $w_t$ is in the vocabulary, we define the probability of selecting copy as:

$$\hat{z}_t = p(1 - z_t | h_t) = \text{sigmoid}(f_{\text{copy}}(\mathbf{h}_t, \mathbf{a}_t)). \tag{4}$$

Here, $f_{\text{copy}}$ is an MLP with one ReLU hidden layer and a single linear output unit. For facts about attributes such as *nationality* or *profession*, the words in the fact description (e.g., "American" or "actor") are likely to be in the vocabulary, but for facts like the *year_of_birth* or *father_name*, the model is likely to choose to copy.

**4) Word Generation**. Word $w_t$ is generated from the source indicated by the copy-switch $\hat{z}_t$ as follows:

$$w_t = \begin{cases} w_t^v \in \mathcal{V}, & \text{if } \hat{z}_t < 0.5, \\ w_t^o \in \mathcal{O}_{a_t}, & \text{otherwise.} \end{cases}$$

For vocabulary word $w_t^v \in \mathcal{V}$, we use the softmax function where each output dimension corresponds to a word in the vocabulary including UNK,

$$P(w_t^v = w | h_t) = \frac{\exp(\mathbf{k}_{\text{voca}}^\top \mathbf{W}[w])}{\sum_{w' \in \mathcal{V}} \exp(\mathbf{k}_{\text{voca}}^\top \mathbf{W}[w'])}. \tag{5}$$

where $\mathbf{k}_{\text{voca}} \in \mathbb{R}^{D_w}$ is obtained by $f_{\text{voca}}(\mathbf{h}_t, \mathbf{a}_t)$ which is an MLP with a ReLU hidden layer and linear output units of dimension $D_w$.

For knowledge word $w_t^o \in \mathcal{O}_{a_t}$, we predict the *position* of the word in the fact description and then copy the word on the predicted position to output. This is because, unlike with the traditional copy mechanism, our context words (i.e., the fact description) often consist of all unknown words and/or are short in length. Copying allows us not to rely on the word embeddings for the knowledge words. Instead, we learn the position embeddings shared among all knowledge words. This makes sense because words in the fact description usually appear one by one in increasing order. Thus, given that the first symbol $o_1 =$ *"Michelle"* was used in the previous time step and prior to that other words such as *"President"* and *"US"* were also observed, the model can easily predict that it is time to select the second symbol, i.e., $o_2 =$ *"Obama"*.

For this copy-by-position, we first generate the position key $\mathbf{k}_{\text{pos}} \in \mathbb{R}^{D_o}$ by a function $f_{\text{poskey}}(\mathbf{h}_t, \mathbf{a}_t)$ which is again an MLP with one hidden layer and linear outputs whose dimension is equal to the maximum length of the fact descriptions $N_{\text{max}}^o = \max_{a \in \mathcal{F}} |O_a|$ where $\mathcal{F} = \cup_k \mathcal{F}_k$. Then, the $n$-th symbol $o_n \in \mathcal{O}_{a_t}$ is chosen by

$$P(w_t^o = o_n | h_t, a_t) = \frac{\exp(\mathbf{k}_{\text{pos}}^\top \mathbf{P}[n])}{\sum_{n'} \exp(\mathbf{k}_{\text{pos}}^\top \mathbf{P}[n'])}, \tag{6}$$

with $n'$ running from 0 to $|\mathcal{O}_{a_t}| - 1$. Here, $\mathbf{P}^{D_o \times N_{\text{max}}^o}$ is the position embedding matrix. Note that $N_{\text{max}}^o$ is typically a much smaller number (e.g., 20 in our experiments) than the size of vocabulary. The position embedding matrix $\mathbf{P}$ is learned during training.

Although in this paper we find that the simple position prediction performs well, we note that one could also consider a more advanced encoding such as one based on a convolutional network (Kim, 2014) to model the fact description. At test time, to compute $p(w_t^k | w_{<t}^k)$, we can obtain $\{z_{<t}^k, a_{<t}^k\}$ from $\{w_{<t}^k\}$ and $\mathcal{F}_k$ using the automatic labeling script, and perform the above inference process with hard decisions taken about $z_t$ and $a_t$ based on the model's predictions.

### 3.3 LEARNING

Given word observations $\{W_k\}_{k=1}^K$ and knowledge $\{\mathcal{F}_k\}_{k=1}^K$, our objective is to maximize the log-likelihood of the observed words w.r.t the model parameter $\theta$,

$$\theta^* = \underset{\theta}{\text{argmax}} \sum_k \log P_\theta(W_k | \mathcal{F}_k). \tag{7}$$

| # topics | # tokens | # unique tokens | # facts | # entities |
|----------|----------|-----------------|---------|------------|
| 10K | 1.5M | 78k | 813k | 560K |

| # relations | $\max_k |\mathcal{F}_k|$ | $\mathrm{avg}_k |\mathcal{F}_k|$ | $\max_a |O_a|$ | $\mathrm{avg}_a |O_a|$ |
|-------------|-----------|-----------|-----------|-----------|
| 1.5K | 1K | 79 | 19 | 2.15 |

Table 1: Statistics of the WikiFacts-FilmActor-v0.1 Dataset.

Because, given $W_k$ and $\mathcal{F}_k$, a sequence of $Y_k = \{y_t = (w_t, z_t, a_t)\}_{t=1:|W_k|}$ is deterministically induced for each word $w_t$, the following equality is satisfied

$$P_\theta(W_k|\mathcal{F}_k) = P_\theta(Y_k|\mathcal{F}_k). \tag{8}$$

By the chain rule, we can decompose the probability of the observation $Y_k$ as

$$\log P_\theta(Y_k|\mathcal{F}_k) = \sum_{t=1}^{|Y_k|} \log P_\theta(y_t^k|y_{1:t-1}^k, \mathcal{F}_k). \tag{9}$$

Then, after omitting $\mathcal{F}_k$ and $k$ for simplicity, we can rewrite the single step conditional probability as

$$P_\theta(y_t|y_{1:t-1}) = P_\theta(w_t, a_t, z_t|h_t) = P_\theta(w_t|a_t, z_t, h_t)P_\theta(a_t|h_t)P_\theta(z_t|h_t). \tag{10}$$

We maximize the above objective using stochastic gradient optimization.

## 4 EVALUATION

### 4.1 WIKIFACTS DATASET

An obstacle in developing the above model is the lack of the dataset where the text corpus is aligned with facts at the word level. To this end, we produced the *WikiFacts* dataset by aligning Wikipedia descriptions with corresponding Freebase facts. Because many Freebase topics provide a link to its corresponding topic in Wikipedia, we choose a set of topics for which both a Freebase entity and a Wikipedia description exist. In the experiments, we used a version called `WikiFacts-FilmActor-v0.1` where the domain is restricted to the */Film/Actor* in Freebase.

For all object entity descriptions $\{O_{a^k}\}$ associated with $\mathcal{F}_k$, we performed string matching to the Wikipedia description $W_k$. We used the summary part (first few paragraphs) of the Wikipedia page as text to be modeled but discarded topics for which the number of facts is greater than 1000 or the Wikipedia description is too short ($< 3$ sentences). For the string matching, we also used the synonyms and alias provided by WordNet (Miller, 1995) and Freebase.

We augmented the fact set $\mathcal{F}_k$ with the *anchor* facts $\mathcal{A}_k$ whose relationship is all set to `UnknownRelation`. That is, observing that an anchor (words under hyperlink) in Wikipedia descriptions has a corresponding Freebase entity as well as being semantically closely related to the topic in which the anchor is found, we make a synthetic fact of the form (*Topic*, `UnknownRelation`, *Anchor*). This potentially compensates for some missing facts in Freebase. Because we extract the anchor facts from the full Wikipedia page and they all share the same relation, it is more challenging for the model to use these anchor facts than using the Freebase facts. As a result, for each word $w$ in the dataset, we have a tuple $(w, z_w, a_w, k_w)$. Here, $k_w$ is the topic where $w$ appears. We provide a summary of the dataset statistics in Table 1. The dataset will be available on a public webpage[4].

### 4.2 EXPERIMENTS

**Setup.** We split the dataset into 80/10/10 for train, validation, and test. As a baseline model, we use the RNNLM. For both the NKLM and the RNNLM, two-layer LSTMs with dropout regularization (Zaremba et al., 2014) are used. We tested models with different numbers of LSTM hidden units [200, 500, 1000], and report results from the 1000 hidden-unit model. For the NKLM, we set the symbol embedding dimension to 40 and word embedding dimension to 400. Under this setting, the number of parameters in the NKLM is slightly smaller than that of the RNNLM. We used

---

[4]`https://bitbucket.org/skaasj/wikifact_filmactor`

| Model | Validation | | | Test | | | # UNK |
|---|---|---|---|---|---|---|---|
| | PPL | UPP | UPP-f | PPL | UPP | UPP-f | |
| RNNLM | 39.4 | 97.9 | 56.8 | 39.4 | 107.0 | 58.4 | 23247 |
| **NKLM** | **27.5** | **45.4** | **33.5** | **28.0** | **48.7** | **34.6** | **12523** |
| no-copy | 38.4 | 93.5 | 54.9 | 38.3 | 102.1 | 56.4 | 29756 |
| no-fact-no-copy | 40.5 | 98.8 | 58.0 | 40.3 | 107.4 | 59.3 | 32671 |
| no-TransE | 48.9 | 80.7 | 59.6 | 49.3 | 85.8 | 61.0 | 13903 |

Table 2: **We compare four different versions of the NKLM to the RNNLM on three different perplexity metrics.** We used 10K vocabulary. In **no-copy**, we disabled the knowledge-copy functionality, and in **no-fact-no-copy**, using topic knowledge is also additionally disabled by setting all facts as NaF. Thus, **no-fact-no-copy** is very similar to RNNLM. In **no-TransE**, we used random vectors instead of the TransE embeddings to initialize the KG entities. As shown, the NKLM shows best performance in all cases. The **no-fact-no-copy** performs similar to the RNNLM as expected (slightly worse partly because it has smaller model parameters than that of the RNNLM). As expected, **no-copy** performs better than **no-fact-no-copy** by using additional information from the fact embedding, but without the copy mechanism. In the comparison of the NKLM and **no-copy**, we can see the significant gain of using the copy mechanism to predict named entities. In the last column, we can also see that, with the copy mechanism, the number of predicting unknown decreases significantly. Lastly, we can see that the TransE embedding is important.

100-dimension TransE embeddings for Freebase entities and relations, and concatenate the relation and object embeddings to obtain fact embeddings. We averaged all fact embeddings in $\mathcal{F}_k$ to obtain the topic context embedding $\mathbf{e}_k$. We unrolled the LSTMs for 30 steps and used minibatch size 20. We trained the models using stochastic gradient ascent with gradient clipping range [-5,5]. The initial learning rate was set to 0.5 for the NKLM and 1.5 for the RNNLM, and decayed after every epoch by a factor of 0.98. We trained for 50 epochs and report the results chosen by the best validation set results.

**Evaluation metric.** The perplexity $\exp(-\frac{1}{N}\sum_{i=1}^{N}\log p_{w_i})$ is the standard performance metric for language modeling. This, however, has a problem in evaluating language models for a corpus containing many named entities: *a model can get good perplexity by accurately predicting UNK words*. As an extreme example, when all words in a sentence are unknown words, a model predicting everything as UNK will get a good perplexity. Considering that unknown words provide virtually no useful information, this is clearly a problem in tasks such as question answering, dialogue modeling, and knowledge language modeling.

To this end, we introduce a new evaluation metric, called the Unknown-Penalized Perplexity (UPP), and evaluate the models on this metric as well as the standard perplexity (PPL). Because the actual word underlying the UNK should be one of the out-of-vocabulary (OOV) words, in UPP, we penalize the likelihood of unknown words as follows:

$$P_{\text{UPP}}(w_{\text{unk}}) = P(w_{\text{unk}})/|\mathcal{V}_{\text{total}} \setminus \mathcal{V}_{\text{voca}}|.$$

Here, $\mathcal{V}_{\text{total}}$ is a set of all unique words in the corpus, and $\mathcal{V}_{\text{voca}}$ is the vocabulary used in the softmax. In other words, in UPP we assume that the OOV set is equal to $|\mathcal{V}_{\text{total}} \setminus \mathcal{V}_{\text{voca}}|$ and thus assign a uniform probability to OOV words. In another version, UPP-fact, we consider the fact that the RNNLM can also use the knowledge given to the NKLM to some extent, but with limited capability (because the model is not designed for it). For this, we assume that the OOV set is equal to the total knowledge vocabulary of a topic $k$, i.e.,

$$P_{\text{UPP-fact}}(w_{\text{unk}}) = P(w_{\text{unk}})/|\mathcal{O}_k|,$$

where $\mathcal{O}_k = \cup_i O_{a^{k,i}}$. In other words, by using UPP-fact, we assume that, for an unknown word, the RNNLM can pick one of the knowledge words with uniform probability. We describe the detail results and discussion on the experiments in the captions of Table 2, 3, and 4.

**Observations from the experiment results.** Our observations from the experiment results are as follows. (a) The NKLM outperforms the RNNLM in all three perplexity measures. (b) The copy mechanism is the key of the significant performance improvement. Without the copy mechanism, the NKLM still performs better than the RNNLM due to its usage of the fact information, but the improvement is not so significant. (c) The NKLM results in a much smaller number of UNKs (roughly, a half of the RNNLM). (d) When no knowledge is available, the NKLM performs as well as the

| Model | Validation | | | Test | | | # UNK |
|---|---|---|---|---|---|---|---|
| | PPL | UPP | UPP-f | PPL | UPP | UPP-f | |
| NKLM_5k | **22.8** | **48.5** | **30.7** | **23.2** | **52.0** | **31.7** | **19557** |
| RNNLM_5k | 27.4 | 108.5 | 47.6 | 27.5 | 118.3 | 48.9 | 34994 |
| NKLM_10k | **27.5** | **45.4** | **33.5** | **28.0** | **48.7** | **34.6** | **12523** |
| RNNLM_10k | 39.4 | 97.9 | 56.8 | 39.4 | 107.0 | 58.4 | 23247 |
| NKLM_20k | **33.4** | **45.9** | **37.9** | **34.7** | **49.2** | **39.7** | **9677** |
| RNNLM_20k | 57.9 | 99.5 | 72.1 | 59.3 | 108.3 | 75.5 | 13773 |
| NKLM_40k | **41.4** | **49.0** | **44.4** | **43.6** | **52.7** | **47.1** | **5809** |
| RNNLM_40k | 82.4 | 107.9 | 92.3 | 86.4 | 116.9 | 97.9 | 9009 |

Table 3: **The NKLM and the RNNLM are compared for vocabularies of four different sizes [5K, 10K, 20K, 40K]**. As shown, in all cases the NKLM significantly outperforms the RNNLM. Interestingly, for the standard perplexity (PPL), the gap between the two models increases as the vocabulary size increases while for UPP the gap stays at a similar level regardless of the vocabulary size. This tells us that the standard perplexity is significantly affected by the UNK predictions, because with UPP the contribution of UNK predictions to the total perplexity is very small. Also, from the UPP value for the RNNLM, we can see that it initially improves when vocabulary size is increased as it can cover more words, but decreases back when the vocabulary size is largest (40K) because the rare words are added last to the vocabulary.

| Warm-up | Louise Allbritton ( 3 july <unk>february 1979 ) was |
|---|---|
| RNNLM | a <unk><unk>who was born in <unk>, <unk>, <unk>, <unk>, <unk>, <unk>, <unk> |
| NKLM | an english [Actor]. he was born in [Oklahoma] , and died in [Oklahoma]. he was married to [Charles] [Collingwood] |
| Warm-up | Issa Serge Coelo ( born 1967 ) is a <unk> |
| RNNLM | actor . he is best known for his role as <unk><unk>in the television series <unk>. he also |
| NKLM | [Film] director . he is best known for his role as the <unk><unk>in the film [Un] [taxi] [pour] [Aouzou] |
| Warm-up | Adam wade Gontier is a canadian Musician and Songwriter . |
| RNNLM | she is best known for her role as <unk><unk>on the television series <unk>. she has also appeared |
| NKLM | he is best known for his work with the band [Three] [Days] [Grace] . he is the founder of the |
| Warm-up | Rory Calhoun ( august 8 , 1922  april 28 |
| RNNLM | , 2010 ) was a <unk>actress . she was born in <unk>, <unk>, <unk>. she was |
| NKLM | , 2008 ) was an american [Actor] . he was born in [Los] [Angeles] california . he was born in |

Table 4: **Sampled Descriptions**. Given the warm-up phrases, we generate samples from the NKLM and the RNNLM. We denote the copied knowledge words by [word] and the UNK words by <unk>. Overall, the RNNLM generates many UNKs (we used 10K vocabulary) while the NKLM is capable to generate named entities even if the model has not seen some of the words at all during training. In the first case, we found that the generated symbols (words in []) conform to the facts of the topic (Louise Allbritton) except that she actually died in Mexico, not in Oklahoma. (We found that the place_of_death fact was missing.) While she is an actress, the model generated a word [Actor]. This is because in Freebase, there exists only /profession/actor but no /profession/actress. It is also noteworthy that the NKLM fails to use the gender information provided by facts; the NKLM uses "he" instead of "she" although the fact /gender/female is available. From this, we see that if a fact is not detected (i.e., NaF), the statistical co-occurrence governs the information flow. Similarly, in other samples, the NKLM generates movie titles (Un Taxi Pour Aouzou), band name (Three Days Grace), and place of birth (Los Angeles). In addition, to see the NKLM's ability to adapt to knowledge updates without retraining, we changed the fact /place_of_birth/Oklahoma to /place_of_birth/Chicago and found that the NKLM replaces "Oklahoma" by "Chicago" while keeping other words the same.

RNNLM. (e) KG embedding using TransE is an efficient way to initialize the fact embeddings. (f) The NKLM generates named entities in the provided facts whereas the RNNLM generates many more UNKs. (g) The NKLM shows its ability to adapt immediately to the change of the knowledge. (h) The standard perplexity is significantly affected by the prediction accuracy on the unknown words. Thus, one need carefully consider it as a metric for knowledge-related language models.

## 5 CONCLUSION

In this paper, we presented a novel Neural Knowledge Language Model (NKLM) that brings the symbolic knowledge from a knowledge graph into the expressive power of RNN language models. The

NKLM significantly outperforms the RNNLM in terms of perplexity and generates named entities which are not observed during training, as well as immediately adapting to changes in knowledge. We believe that the WikiFact dataset introduced in this paper, can be useful in other knowledge-related language tasks as well. In addition, the Unknown-Penalized Perplexity introduced in this paper in order to resolve the limitation of the standard perplexity, can be useful in evaluating other language tasks. The task that we investigated in this paper is limited in the sense that we assume that the true topic of a given description is known. Relaxing this assumption by making the model search for proper topics on-the-fly will make the model more practical. We believe that there are many more open research challenges related to the knowledge language models.

## ACKNOWLEDGMENTS

The authors would like to thank Alberto García-Durán, Caglar Gulcehre, Chinnadhurai Sankar, Iulian Serban and Sarath Chandar for feedback and discussions as well as the developers of Theano (Bastien et al., 2012), NSERC, CIFAR, Samsung and Canada Research Chairs for funding, and Compute Canada for computing resources.

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

APPENDIX: HEATMAPS

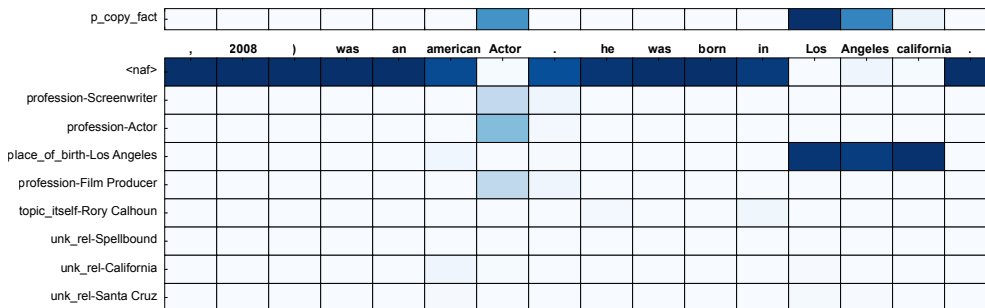

Figure 2: This is a heatmap of an example sentence generated by the NKLM having a warmup *"Rory Calhoun ( august 8 , 1922 april 28"*. The first row shows the probability of knowledge-copy switch (Equation 5 in Section 3.1). The bottom heat map shows the state of the topic-memory at each time step (Equation 2 in Section 3.1). In particular, this topic has 8 facts and an additional <NaF> fact. For the first six time steps, the model retrieves <NaF> from the knowledge memory, copy-switch is off and the words are generated from the general vocabulary. For the next time step, the model gives higher probability to three different profession facts: "Screenwriter", "Actor" and "Film Producer." The fact "Actor" has the highest probability, copy-switch is higher than 0.5, and therefore "Actor" is copied as the next word. Moreover, we see that the model correctly retrieves the place of birth fact and outputs "Los Angeles." After that, the model still predicts the place of birth fact, but copy-switch decides that the next word should come from the general vocabulary, and outputs "California."

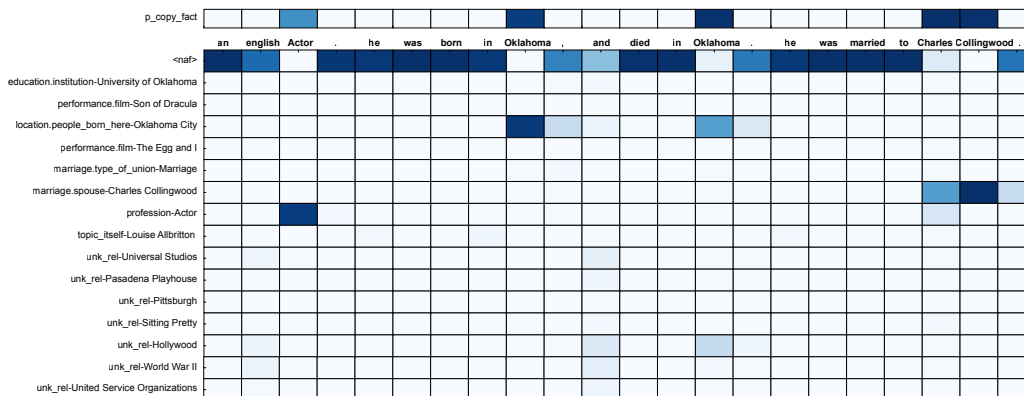

Figure 3: This is an example sentence generated by the NKLM having a warmup *"Louise Allbritton ( 3 july <unk>february 1979 ) was"*. We see that the model correctly retrieves and outputs the profession ("Actor"), place of birth ("Oklahoma"), and spouse ("Charles Collingwood") facts. However, the model makes a mistake by retrieving the place of birth fact in a place where the place of death fact is supposed to be used. This is probably because the place of death fact is missing in this topic memory and then the model searches for a fact about location, which is somewhat encoded in the place of birth fact. In addition, *Louise Allbritton* was a woman, but the model generates a male profession "Actor" and male pronoun "he". The "Actor" is generated because there is no "Actress" representation in Freebase.

