# Peer review of "A Neural Knowledge Language Model"

_ICLR 2017 — rejected_

[Official Review · AnonReviewer3 · rating 6 · confidence 3 · 18 Dec 2016]
**No Title**

This paper addresses the practical problem of generating rare or unseen words in the context of language modeling. Since language follows a Zipf’s law, most approaches limit the vocabulary (because of computation reasons) and hence rare words are often mapped to a UNK token. Rare words are especially important in context of applications such as question answering. MT etc. This paper proposes a language modeling technique which incorporates facts from knowledge bases (KBs) and thus has the ability to generate (potentially unseen) words from KBs. This paper also releases a dataset by aligning words with Freebase facts and corresponding Wikipedia descriptions.

The model first selects a KB fact based on the previously generated words and facts. Based on the selected fact, it then predicts whether to generate a word based on the vocabulary or to output a symbolic word from the KB. For the latter, the model is trained to predict the position of the word from the fact description.

Overall the paper could use some rewriting especially the notations in section 3. The experiments are well executed and they definitely get good results. The heat maps at the end are very insightful. 

Comments

This contributions of this paper would be much stronger if it showed improvements in a practical applications such as Question Answering (although the paper clearly mentions that this technique could be applied to improve QA)
In section 3, it is unclear why the authors refer the entity as a ‘topic'. This makes the text a little confusing since a topic can also be associated with something abstract, but in this case the topic is always a freebase entity. 
Is it really necessary to predict a fact at every step before generating a word. In other words, how many distinct facts on average does the model choose to generate a sentence. Intuitively a natural language sentence would be describe few facts about an entity. If the fact generation step could be avoided (by adding a latent variable which decides if the fact should be generated or not), the model will also be faster.
In equation 2, the model has to make a hard decision to choose the fact. For this to be end to end trained, every word needs to be annotated with a corresponding fact which might not be always a realistic scenario. For e.g., in domains such as social media text.
Learning position embeddings for copying knowledge words seems a little counter-intuitive. Does the sequence of knowledge words follow any particular structure like word O_2 is always the last name (e.g. Obama).
It would also be nice to compare to char-level LM's which inherently solves the unknown token problem.

[Official Review · AnonReviewer2 · rating 6 · confidence 4 · 20 Dec 2016]

This paper proposes to incorporate knowledge base facts into language modeling, thus at each time step, a word is either generated from the full vocabulary or relevant KB entities.

The authors demonstrate the effectiveness on a new generated dataset WikiFacts which aligns Wikipedia articles with Freebase facts.  The authors also suggest a modified perplexity metric which penalizes the likelihood of unknown words.

At a high level, I do like the motivation of this paper -- named entity words are usually important for downstream tasks, but difficult to learn solely based on statistical co-occurrences. The facts encoded in KB could be a great supply for this.

However, I find it difficult to follow the details of the paper (mainly Section 3) and think the paper writing needs to be much improved. 
- I cannot find where  f_{symbkey} / f_{voca} / f_{copy} are defined
- w^v, w^s are confusing.
- e_k seems to be the average of all previous fact embeddings? It is necessary to make it clear enough.
- (h_t, c_t) = f_LSTM(x_{t−1}, h_{t−1})  c_t is not used?
- The notion of “fact embeddings” is also not that clear (I understand that they are taken as the concatenation of relation and entity (object) entities in the end).  For the anchor / “topic-itself” facts, do you learn the embedding for the special relations and use the entity embeddings from TransE?

On generating words from KB entities (fact description), it sounds a bit strange to me to generate a symbol position first.  Most entities are multiple words, and it is necessary to keep that order. Also it might be helpful to incorporate some prior information, for example, it is common to only mention “Obama” for the entity “Barack Obama”?

[Official Review · AnonReviewer1 · rating 6 · confidence 4 · 20 Dec 2016]

The paper proposes an evolution upon traditional Recurrent Language Models to give the capability to deal with unknown words. It is done by pairing the traditional RNNLM with a module operating on a KB and able to copy from KB facts to generate unseen words. It is shown to be efficient and much better than plain RNNLM on a new dataset.

The writing could be improved. The beginning of Section 3 in particular is hard to parse.

There have been similar efforts recently (like "Pointer Sentinel Mixture Models" by Merity et al.) that attempt to overcome limitations of RNNLMs with unknown words; but they usually do it by adding a mechanism to copy from a longer past history. The proposal of the current paper is different and more interesting to me in that it try to bring knowledge from another source (KB) to the language model. This is harder because one needs to leverage the large scale of the KB to do so. Being able to train that conveniently is nice.

The architecture appears sound, but the writing makes it hard to fully understand completely so I can not give a higher rating. 


Other comments:
* How to cope with the dependency on the KB? Freebase is not updated anymore so it is likely that a lot of the new unseen words in the making are not going to be in Freebase.
* What is the performance on standard benchmarks like Penn Tree Bank?
* How long is it to train compare to a standard RNNLM?
* What is the importance of the knowledge context $e$?
* How is initialized the fact embedding $a_{t-1}$ for the first word?
* When a word from a fact description has been chosen as prediction (copied), how is it encoded in the generation history for following predictions if it has no embedding (unknown word)? In other words, what happens if "Michelle" in the example of Section 3.1 is not in the embedding dictionary, when one wants to predict the next word?

[Author Response · Sungjin Ahn · 20 Jan 2017]
**Reminder of the improvements**

Dear Reviewers and AreaChair, 

Approaching to the deadline, I'd like to remind the reviewers of the fact that the revised version of the paper (that we uploaded in Dec. 27) has significant improvements with respect to most of the comments pointed by the reviewers, which are mainly in terms of the writing clarity in Section 3.

[Final Decision · Program Chairs · 06 Feb 2017]
**ICLR committee final decision**

This work introduces a combination of a LM with knowledge based retrieval system. This builds upon the recent trend of incorporating pointers and external information into generation, but includes some novelty, making the paper "different and more interesting". Generally though the reviewers found the clarity of the work to be sufficiently an issue that no one strongly defended its inclusion.
 
 Pros:
 - The reviewers seemed to like the work and particularly the problem space. Issues were mainly on presentation and experiments. 
 
 Mixed:
 - Reviewers were divided on experimental quality. The work does introduce a new dataset, but reviewers would also have liked use on some existing tasks. 
 
 Cons:
 - Clarity and writing issues primarily. All reviewers found details missing and generally struggled with comprehension.
 - Novelty was a question. Impact of work could also be improved by more clearly defining new contributions